# Compute-Optimal Solutions for Acoustic Wave Equation Using Hard-Constraint PINNs

## Abstract

This paper explores the optimal imposition of hard constraints, strategic sampling of PDEs, and computational domain scaling for solving the acoustic wave equation within a specified computational budget. First, we derive a formula to systematically enforce hard boundary and initial conditions in Physics-Informed Neural Networks (PINNs), employing continuous functions within the PINN ansatz to ensure that these conditions are satisfied. We demonstrate that optimally selecting these functions significantly enhances the convergence of the solution. Secondly, we introduce a Dynamic Amplitude-Focused Sampling (DAFS) method that optimizes the efficiency of hard-constraint PINNs under a fixed number of sampling points. Leveraging these strategies, we develop an algorithm to determine the optimal computational domain size, given a computational budget. Our approach offers a practical framework for domain decomposition in large-scale implementation of acoustic wave equation systems.

## 1 Introduction

The concept of using artificial neural networks to solve differential equations was first explored in the 1990s by Lagaris et al. [1998]. In the work of Lagaris et al. [1998], they developed an ansatz solution that inherently satisfies the boundary conditions (BC) and the initial conditions (IC) of differential equations. More recently, the advent of physics-informed neural networks (PINNs) was marked by the influential study of Raissi et al. [2019]. This work leverages modern deep neural networks to solve forward and inverse problems involving nonlinear partial differential equations (PDEs), incorporating BCs and ICs through soft constraints in loss functions.

Subsequent research has introduced various modifications to PINNs to enhance their accuracy, efficiency, and scalability [Lu et al., 2021a]. There are a couple of drawbacks for many PINNs with soft constraints for BCs and ICs. The selection of weights and samples for BCs and ICs cannot certainly be determined and requires many trial-and-error tests. Even when the loss function is minimized, the BCs and ICs are not strictly satisfied. To target the scaling problems of general PDEs and take advantage of parallel computing, XPINNs and FBPINNs have been developed based on domain decomposition methods [Jagtap and Karniadakis, 2020, Shukla et al., 2021, Moseley et al., 2023].

There are a few key points that these previous reseearches missed. First, how to formulate ansatz solutions satisfying BCs and ICs, specifically the function multiplier of NN. Second, if BC and IC are inherently satisfied by constructing the ansatz solution, how to optimally sample the PDEs in the training process. Furthermore, for the existing PINNs handling scaling problems, how to decompose the domain to save the overall compute budget.

Submitted to 38th Conference on Neural Information Processing Systems (NeurIPS 2024). Do not distribute.

In this paper, we set up a 1D wave equation problem and investigate the optimal sampling and constraint imposing method given a compute budget.

The contributions of this paper are as follows.

- We systematically derived the implementation of hard BC and IC constraints in PINNs to solve acoustic wave equations. We give a strategy to select basic functions in the PINN ansatz solution that guarantee the satisfaction of BCs and ICs. We find that optimal selection of the basic function in the PINN ansatz can improve the convergence of PINNs.

- We developed a Dynamic Amplitude-Focused Sampling (DAFS) algorithm to improve the convergence of hard-constraint PINNs for wave equations given a fixed number of sampling points.

- With the hard constraint and importance sampling strategies, we propose an algorithm to find the optimal size of the computational given a compute budget. This domain size optimization algorithm can help the domain decomposition-based PINNs for large-scale problems save computational cost.

## 2   Related Work

**Hard constraint**   Hard constraint PINNs can guarantee the satisfaction of BCs, ICs, symmetries, and/or conservation laws. There are comprehensive studies of embedding BCs in PINNs. Lu et al. [2021b] demontrated various ansatz equations to strictly meet Dirichlet and periodic BCs, and proposed the penalty method and the augamented Lagrangian method to impose inequality constraints as hard constraints. Liu et al. [2022] developed a unified ansatz formula to enforce the Dirichlet, Neumann, and Robin boundary conditions for high-dimensional and geometrically complex domains. Moseley et al. [2023] implemented the hard Dirichlet in the subdomain using a $\tanh^2(\omega x)$ function as the multiplier function of the neural networks in their FBPINN ansatz solution. However, studies on how to impose both hard BC and IC constraints in PINNs for acoustic wave equations that have a second-order time dirivative term are still limited. Alkhadhr and Almekkawy [2023] compared the accuracy and performance of PINNs with a combination of hard-BC/soft-BC and hard-IC/soft-IC for solving a 1D wave equation with a time-dependent point source function. This implementation of the hard-IC only considers the satisfaction of the wavefield values at the initial time $u(x, t = 0)$, but neglects the hard constraint of the first-order time derivative of the wavefield $u(x, t)$, i.e., $\partial_t u(x, t = 0)$. Brecht et al. [2023] proposed improved physics-informed DeepONets with hard constraints, and presented a numerical example of a 1D standing wave equation with Dirichlet BCs. The DeepONet framework used in the paper has an inherent satisfaction of the initial wavefield, but $\partial_t u(x, t = 0)$ is also neglected. This neglection does not affect the numerical results for the 1D standing wave equation in their paper, since they simply assume $\partial_t u(x, t = 0) = 0$.

**Strategic Sampling**   Many sampling algorithms have been developed to improve the training effi-ciency, mitigating failure modes of PINNs. [Wu et al., 2023] provided a comprehensive comparison of ten sampling methods, including non-adaptive and residual-based adaptive methods. Daw et al. [2023] proposed a Retain-Resample-Release (R3) Sampling algorithm to mitigate the failure propagation during the training processes of PINNs. [Gao et al., 2023a,b] developed failure informed adamptive sampling for PINNs, with the extentions of combining re-sampling and subset simulation. Yang et al. [2023] introduced a Dynamic Mesh-Based Importance Sampling (DMIS) method to enhance the training of PINNs. Additionally, [Zhang et al., 2024] proposed an annealed adaptive importance sampling method for solving high-dimensional partial differential equations using PINNs.

**Domain Scaling**   Computational domain scaling is a key issue to apply PINNs to real-world large spatial-temporal scale applications. [Jagtap and Karniadakis, 2020] proposed a generalized space-time domain decomposition framework for PINNs, named extended PINNs (XPINNs), which can handle nonlinear PDEs on complex-geometry domains. XPINNs provide large representation and parallelization capacity by deploying multiple neural networks in smaller subdomains, offering both space and time parallelization to reduce training costs effectively. Shukla et al. [2021] developed a distributed framework for PINNs based on two extensions: conservative PINNs (cPINNs) and XPINNs. These methods employ domain decomposition in space and time-space, respectively, enhancing the parallelization capacity, representation capacity, and efficient hyperparameter tuning of

PINNs. The framework allows for optimizing all hyperparameters of each neural network separately in each subdomain, providing significant advantages for multi-scale and multi-physics problems. They demonstrated the efficiency of cPINNs and XPINNs through various forward problems, highlighting that cPINNs are more communication-efficient while XPINNs offer greater flexibility for handling complex subdomains. Moseley et al. [2023] addressed the limitations of PINNs in solving large domains and multi-scale solutions by proposing Finite Basis PINNs (FBPINNs). FBPINNs use neural networks to learn basis functions defined over small, overlapping subdomains, inspired by classical finite element methods. This approach mitigates the spectral bias of neural networks and reduces the complexity of the optimization problem by using smaller neural networks in a parallel, divide-and-conquer approach. Their experiments showed that FBPINNs outperform standard PINNs in accuracy and computational efficiency for both small and large, multi-scale problems. Chalapathi et al. [2024] introduced a scalable approach to enforce hard physical constraints using Mixture-of-Experts (MoE) in neural network architectures. This method imposes constraints over smaller decomposed domains, with each domain solved by an expert through differentiable optimization. The independence of each expert allows for parallelization across multiple GPUs, improving accuracy, training stability, and computational efficiency for predicting the dynamics of complex nonlinear systems. The optimal decomposition of subdomains is critical to the effectiveness of these scaling methods, given a fixed compute budget. Our work focuses on finding the maximum subdomain size that even a 64x2 small PINN can handle within a compute budget.

# 3 Methodology

In this section, we outline our approach to effectively implement hard constraints, strategically sampling partial differential equations (PDEs), and optimizing the scaling of computational domains. These methods are utilized to solve the acoustic wave equation within a specified computational budget.

We focus on an acoustic wave equation defined by:

$$
\begin{aligned}
\mathcal{D}[\mathbf{u}(\mathbf{x},t); c(\mathbf{x})] &= f(\mathbf{x},t), & \mathbf{x} \in \Omega, \quad t \in [t_0, T], \\
\mathcal{B}_i[\mathbf{u}(\mathbf{x},t)] &= U_i(\mathbf{x},t), & \mathbf{x} \in \partial\Omega_i, \quad t \in [t_0, T], \\
\mathcal{I}_j[\mathbf{u}(\mathbf{x},t_0)] &= V_j(\mathbf{x}), & \mathbf{x} \in \Omega,
\end{aligned}
\tag{1}
$$

where:

- $\mathcal{D}$ represents the differential operator. For a simplified one-dimensional acoustic wave equation, $\mathcal{D} = \partial_{tt} - c^2(\mathbf{x})\nabla^2$, indicating the second temporal derivative minus the spatial derivative scaled by the square of the local speed of sound, $c(\mathbf{x})$.
- $\mathcal{B}_i$ denotes the boundary condition operator applied at $\mathbf{x} \in \partial\Omega_i$.
- $\mathcal{I}_j$ signifies the initial condition operator, defining the state of the system at $t = t_0$ across the domain $\Omega$.

## 3.1 Hard constraint imposing

A prevalent ansatz employed in prior studies on hard-constraint PINNs for 1D wave equations is expressed as:

$$
u(x,t) = \tau(t)\tilde{u}(x,t) + (1 - \tau(t))u(x,0),
\tag{2}
$$

where $\tilde{u}(x,t)$ represents the neural network output with inputs $x$ and $t$, and $\tau(t)$ is a function that satisfies $\tau(0) = 0$. This design ensures that the initial condition $u(x,0)$ is met precisely when $t = 0$.

To accommodate boundary conditions (BCs) at $x = 0$ and $x = L$, the ansatz is often modified to:

$$
u(x,t) = x(L - x)\tilde{u}(x,t) + U_i(x,t),
\tag{3}
$$

ensuring that $u(x_i, t) = U_i(x_i, t)$ for $x \in \partial\Omega_i$.

A more comprehensive form,

$$
\begin{aligned}
u(x,t) = & x(L-x)\tau(t)\tilde{u}(x,t) + (1 - \tau(t))u(x,0) \\
& + \frac{L-x}{L}(u(0,t) - (1 - \tau(t))u(0,0)) \\
& + \frac{x}{L}(u(L,t) - (1 - \tau(t))u(L,0)),
\end{aligned}
\tag{4}
$$

127 can ensure both Dirichlet BCs and the initial condition $u(x,t)|_{t=0} = u(x,0)$. However, this ansatz
128 does not account for $\partial_t u(x,t)|_{t=0}$, unless it is assumed to be zero.

129 We propose a more general hard constraint imposition formula:

$$
\begin{aligned}
u(x,t) =&\, x(L-x)\tau(t)\tilde{u}(x,t) + ((1-\tau(t)) + t\partial_t)u(x,0) \\
&+ \frac{L-x}{L}(u(0,t) - ((1-\tau(t)) + t\partial_t)u(0,0)) \\
&+ \frac{x}{L}(u(L,t) - ((1-\tau(t)) + t\partial_t)u(L,0)),
\end{aligned}
\tag{5}
$$

130 which guarantees satisfaction of the conditions:

$$
\begin{aligned}
u(x,t) &= U_i(x,t), & x &\in \partial\Omega_i, \\
u(x,t)|_{t=0} &= V_j(x), & x &\in \Omega, \\
\partial_t u(x,t)|_{t=0} &= W_j(x), & x &\in \Omega,
\end{aligned}
\tag{6}
$$

131 where $U_i(x,t)$, $V_j(x)$, $W_j(x)$ are the specified functions in BCs and ICs, and $\tau(t)$ is an arbitrary
132 function satisfying $\tau(0) = d_t\tau(0) = 0$.

133 It is straightforward to demonstrate that the proposed ansatz correctly imposes all BCs and ICs as
134 required:

$$
\begin{cases}
u(x,t)|_{x=0} &= u(0,t), \\
u(x,t)|_{x=L} &= u(L,t), \\
u(x,t)|_{t=0} &= u(x,0), \\
\partial_t u(x,t)|_{t=0} &= \partial_t u(x,0).
\end{cases}
\tag{7}
$$

135 In Section 4.2, we will explore numerical tests to optimize the selection of $\tau(t)$ by evaluating
136 convergence rates and mean absolute errors (MAE).

137 The primary advantage of employing hard constraints in our model is the elimination of the need to
138 fine-tune the weights of PDE, BC, and IC loss terms typically required in soft-constraint PINNs.

## 3.2 Sampling strategy

140 Sampling is crucial for efficient training of PINNs, ensuring rapid convergence and mitigating
141 potential failure modes. To enhance the computational efficiency of our hard-constraint PINNs,
142 we introduce the Dynamic Amplitude-Focused Sampling (DAFS) method. This strategy optimally
143 selects the number of points, $N_{pde}$, used in the training.

144 Initially, we segmented the computational domain to identify regions with high-amplitude acoustic
145 wave fields, based on low-resolution finite difference (FD) simulations. These high-amplitude regions
146 are defined by a threshold $\delta$, which determines the intensity level above which areas are considered
147 to be of high amplitude. Within these identified regions, we uniformly sampled $\alpha N_{pde}$ points. This
148 was supplemented by uniformly sampling $(1-\alpha)N_{pde}$ points in the remaining areas of the domain.

149 Both  and $\alpha$ are parameters crucial to the sampling process and are optimally chosen to balance the
150 computational budget and the accuracy of the simulations. By adjusting these parameters, we can
151 tailor the distribution of sample points to areas that are most influential in the wave dynamics, thereby
152 improving the efficiency of our PINN training.

153 The pseudocode for the DAFS algorithm is provided in Algorithm 1.

154 This sampling strategy, characterized by its focus on dynamically identified regions of interest based
155 on wave amplitude, significantly optimizes the efficiency of the computation during the PINN training
156 phase. The numerical tests for DAFS are in Section 4.3.

# 4 Experiments

## 4.1 Problem setup

159 We applied our method to three numerical examples for three different types of 1D acoustic wave
160 equations — standing waves, string waves, and traveling waves. The ground truth wavefields are
161 shown in Figure 1.

**Algorithm 1** Dynamic Amplitude-Focused Sampling (DAFS)

---

**Require:** $N_{\text{pde}}, \alpha$, domain, FD results (low-resolution Finite Difference results indicating amplitude)
**Ensure:** Sampled points for training
  1: Initialize points $\leftarrow []$
  2: Identify high-amplitude regions from FD results
  3: $N_{\text{high}} \leftarrow \alpha N_{\text{pde}}$         $\triangleright$ Number of points in high-amplitude regions
  4: $N_{\text{low}} \leftarrow (1 - \alpha) N_{\text{pde}}$         $\triangleright$ Number of points in low-amplitude regions
  5: Uniformly sample $N_{\text{high}}$ points in high-amplitude regions and add to points
  6: Uniformly sample $N_{\text{low}}$ points in the remaining areas of the domain and add to points
        **return** points

---

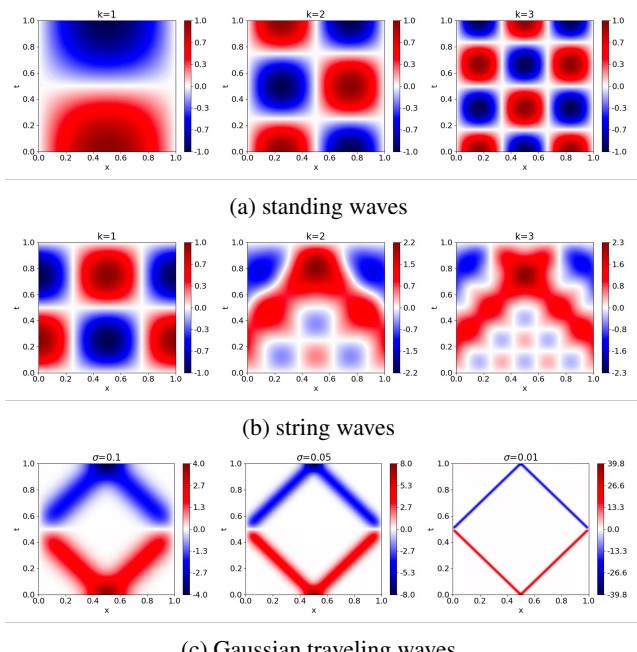

(a) standing waves

(b) string waves

(c) Gaussian traveling waves

Figure 1: Ground truth wavefields for (a) standing waves, (b) string waves, and (c) traveling waves with $k = 1, 2, 3$.

**Standing waves for Dirichlet BCs**   Our first numerical example is a standing wave solution for the following 1D wave equation with Dirichlet BCs:

$$\frac{\partial^2 u(x,t)}{\partial t^2} - c^2 \frac{\partial^2 u}{\partial x^2} = 0, \ x \in (0, L)$$

$$\textbf{B.C.:} \ u(0,t) = u(L,t) = 0,$$

$$\textbf{I.C.:} \ u(x,0) = U(x), \ \frac{\partial u}{\partial t}(x,0) = V(x). \tag{8}$$

The analytical solution $u(x,t)$ for Equation 8 is

$$u(x,t) = \sum_{n=1}^{\infty} A_n \sin\left(\frac{n\pi x}{L}\right) \cos\left(\frac{n\pi c t}{L}\right) + B_n \sin\left(\frac{n\pi x}{L}\right) \sin\left(\frac{n\pi c t}{L}\right). \tag{9}$$

A standing wave solution

$$u(x,t) = \sin\left(\frac{k\pi x}{L}\right) \cos\left(\frac{k\pi c t}{L}\right), k \in \mathbb{Z}^+ \tag{10}$$

can be achieved if we assume $U(x) = \sin\left(\frac{k\pi x}{L}\right)$ and $V(x) = 0$. We show the solutions for $k = 1, 2, 3$ in Figure 1(a).

**String waves for time-dependent BCs** Our third example is a string wave solution for time-dependent BCs shown in Equation 11. The ground truth solutions in Figuer 1(b) are achieved by finite different simulation.

$$\frac{\partial^2 u(x,t)}{\partial t^2} - c^2 \frac{\partial^2 u}{\partial x^2} = 0,\ x \in (0, L)$$
$$\textbf{B.C.:}\ u(0,t) = u(L,t) = \sin(2\pi t),$$
$$\textbf{I.C.:}\ u(x,0) = 0,\ \frac{\partial u}{\partial t}(x,0) = 2\pi \cos\left(\frac{2k\pi x}{L}\right)$$

(11)

**Traveling waves for Gaussian source time functions** Our third example is a traveling wave solution for initial conditions of Gaussian source time functions shown in Equation 12. The ground truth solutions in Figuer 1(c) are computed by finite different simulation.

$$\frac{\partial^2 u(x,t)}{\partial t^2} - c^2 \frac{\partial^2 u}{\partial x^2} = 0,\ x \in (0, L)$$
$$\textbf{B.C.:}\ u(0,t) = u(L,t) = 0,$$
$$\textbf{I.C.:}\ u(x,0) = \frac{1}{\sigma\sqrt{2\pi}} \exp\left(-\frac{(x-\mu)^2}{2\sigma^2}\right),\ \frac{\partial u}{\partial t}(x,0) = 0$$

(12)

### 4.2 Optimal $\tau(t)$ selection for hard constraints

We selected six candidate functions for $\tau(t)$ to construct PINNs with a network configuration of only 64x2 neurons. Figures 2 through 4 illustrate the $L^2$ loss and $L^1$ error as functions of training epochs. Our findings suggest that $\tau(t)$ significantly influences both the convergence rate and the emergence of failure modes. In general, $t^2$, $\frac{2t^2}{1+t^2}$ performs better in general, especially for higher modes $k = 2, 3$. We show a few training dynmaics in Appendix C.

Our analysis indicates that the frequency characteristics of $\tau(t)$ and the corresponding wavefields may be critical for selecting an appropriate $\tau(t)$. Matching these characteristics can potentially enhance the model's efficiency by aligning $\tau(t)$'s influence on the neural network's learning dynamics with the physical properties of the wave phenomena being modeled.

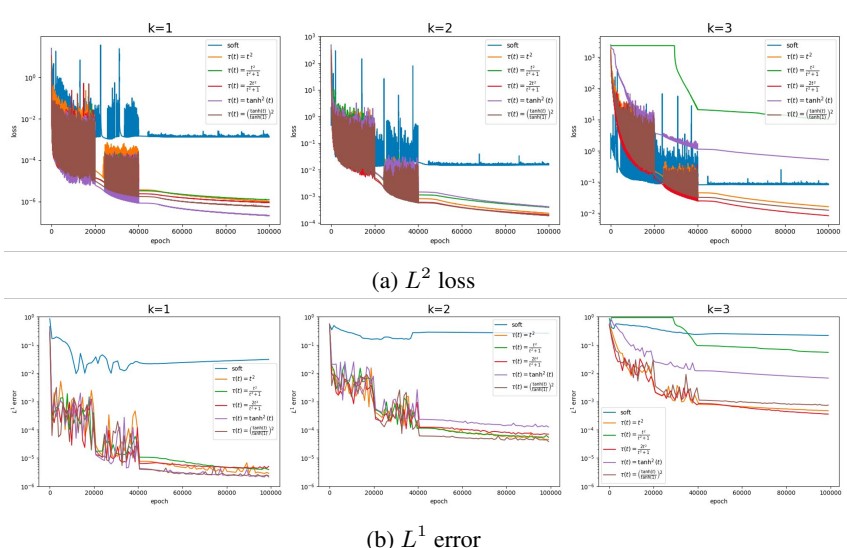

(a) $L^2$ loss

(b) $L^1$ error

Figure 2: $L^2$ loss and $L^1$ error for standing waves with PINNs constructed using six canditate $\tau(t)$ functions.

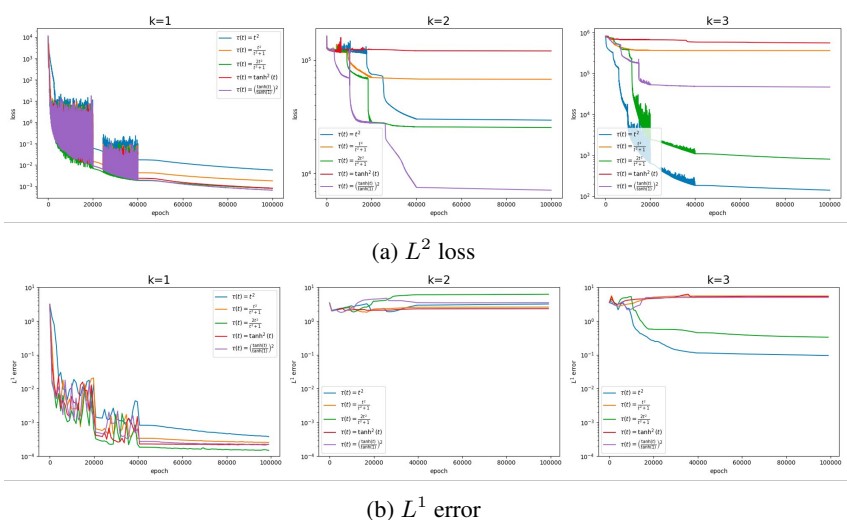

(a) $L^2$ loss

(b) $L^1$ error

Figure 3: $L^2$ loss and $L^1$ error for string waves with PINNs constructed using six canditate $\tau(t)$ functions.

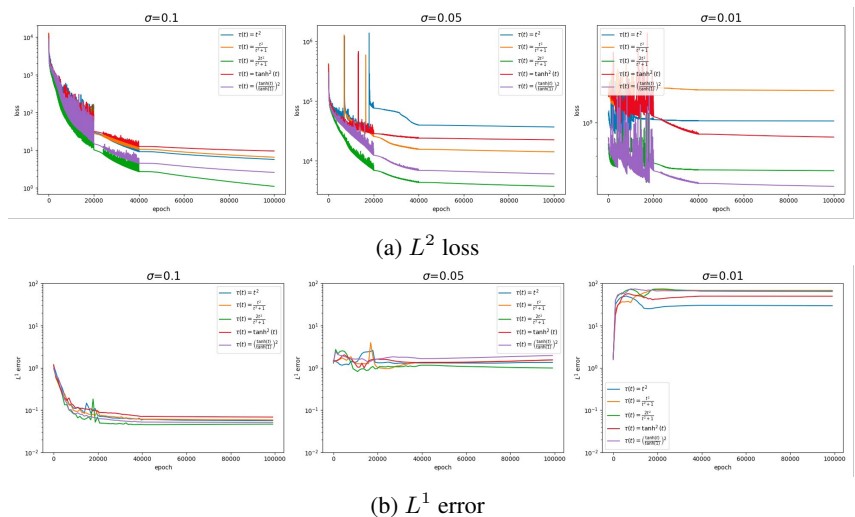

(a) $L^2$ loss

(b) $L^1$ error

Figure 4: $L^2$ loss and $L^1$ error for travelling Gaussian waves with PINNs constructed using six canditate $\tau(t)$ functions.

### 4.3 Dynamic Amplitude-Focused Sampling

We demonstrate the efficacy of our proposed Dynamic Amplitude-Focused Sampling (DAFS) in enhancing both the convergence and accuracy of Physics-Informed Neural Networks (PINNs). Experiments varying $\alpha$ from 0 to 0.5 to 1 indicate that optimal results are typically achieved when $\alpha$ is around 0.5.

This suggests a balanced sampling strategy, where a significant portion of the samples is concentrated in regions of higher amplitude. However, exclusively focusing on these high-amplitude areas can hinder information transfer from boundary conditions to the interior of the domain, potentially leading to failure modes. Figures 5 and 6 illustrate these dynamics, showing the $L^2$ loss and $L^1$ error across different values of $\alpha$, and the impact on the predicted wavefield and its accuracy.

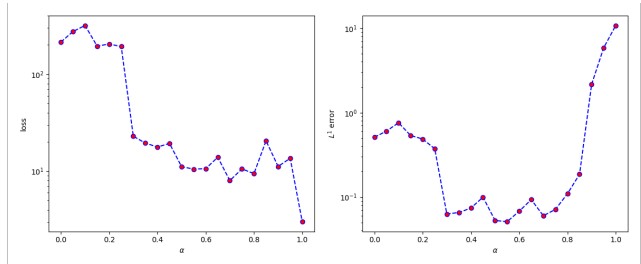

Figure 5: $L^2$ loss and $L^1$ error with varied $\alpha$ from 0 to 1.

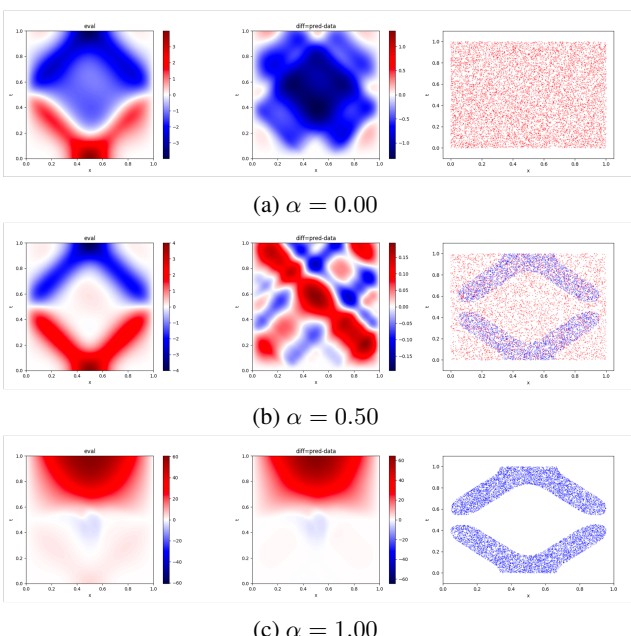

(a) $\alpha = 0.00$

(b) $\alpha = 0.50$

(c) $\alpha = 1.00$

Figure 6: Visualizations for $\alpha = 0.00$, 0.50, and 1.00 (top to bottom): Left - Predicted wavefield, Middle - Difference between the prediction and ground truth, Right - Sampling distribution.

### 4.4 Optimal subdomain

We then propose an optimal subdomain selection method shown in a flow chart in Figure 7. This method will automatically determine the optimal $k$ our 64x2 small PINNs can handle, given a compute budget.

## 5 Limitations and Training Dynamics

While our proposed methods significantly enhance the functionality and efficiency of PINNs, the determination of the optimal function $\tau(t)$ presents certain limitations. The choice of $\tau(t)$ is crucial as it directly affects the model's ability to satisfy boundary and initial conditions rigidly. However, finding an ideal $\tau(t)$ that adapts across different problems and boundary conditions without extensive trial and error remains challenging. The training dynamics are also sensitive to the form of $\tau(t)$, where inappropriate selections can lead to slower convergence or even divergence in some cases. These issues underscore the need for a more automated, perhaps adaptive, approach to selecting $\tau(t)$ that can dynamically adjust based on the evolving training characteristics and the specific requirements of the PDE being solved.

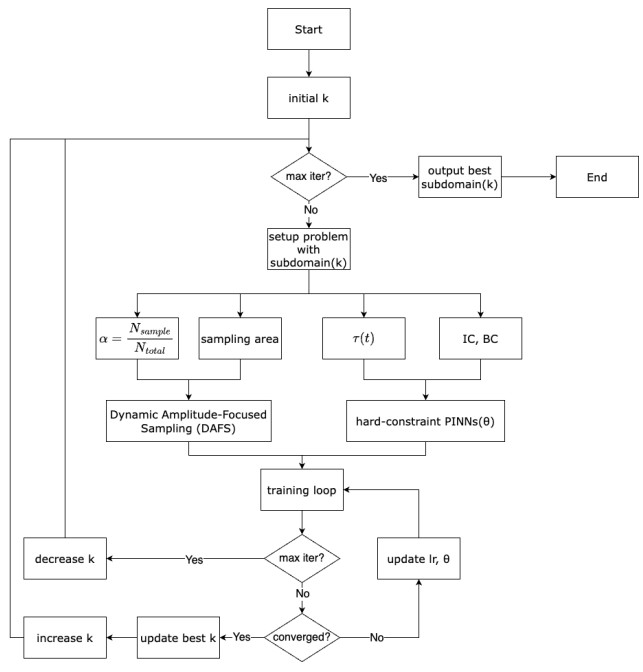

Figure 7: The flow chart of optimal subdomain determination.

## 6 Conclusion

This work presented a comprehensive approach to improving the effectiveness and efficiency of Physics-Informed Neural Networks (PINNs) for solving acoustic wave equations. By integrating a well-formulated hard constraint imposition strategy and the novel Dynamic Amplitude-Focused Sampling (DAFS) method, we have significantly enhanced both the accuracy and convergence of PINNs.

Our methodological innovations include:

- A systematic derivation of hard boundary and initial conditions in PINNs that ensures these constraints are inherently satisfied, leading to better convergence and stability of the solution.

- The introduction of DAFS, which optimally allocates computational resources by focusing sampling in regions of high amplitude while ensuring adequate coverage across the computational domain to prevent information isolation.

- Development of a domain size optimization algorithm that assists in domain decomposition, enabling efficient scaling of PINNs for large-scale applications while managing computational costs.

These contributions mark a significant step forward in the practical deployment of PINNs, especially in fields requiring the simulation of complex physical phenomena over large scales. Future work will focus on extending these strategies to other types of partial differential equations and exploring the integration of our methods with other deep learning frameworks to further enhance the adaptability and efficiency of PINNs in diverse applications, for example, we will explore the integration of our methods with existing PINNs frameworks that employ domain decomposition techniques, such as XPINNs and FBPINNs, to further enhance their scalability and adaptability. We aim to make PINNs more adaptable and efficient for a broader range of applications, particularly in complex systems where traditional numerical methods struggle. By advancing these strategies, we can significantly contribute to the deployment of PINNs in real-world scenarios, tackling large-scale and multi-scale challenges effectively.

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

## A  Phase diagrams of loss weights

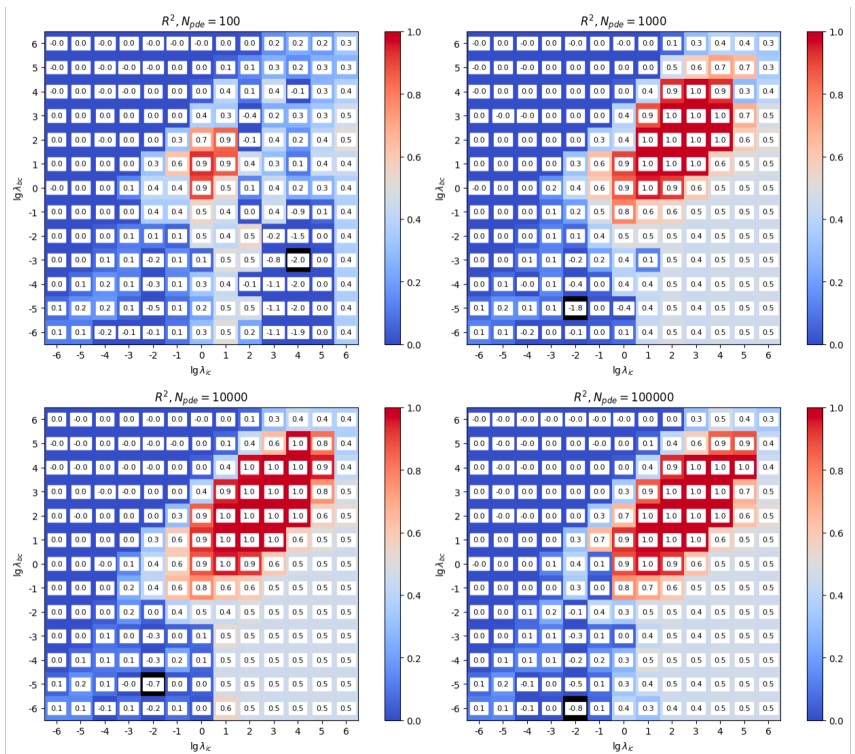

Figure 8: Phase diagrams

## B  Seed

## C  Training dynmaics

mono:

string: increase Npde to $10^4$, we have converged solution(each $10^4$ steps):

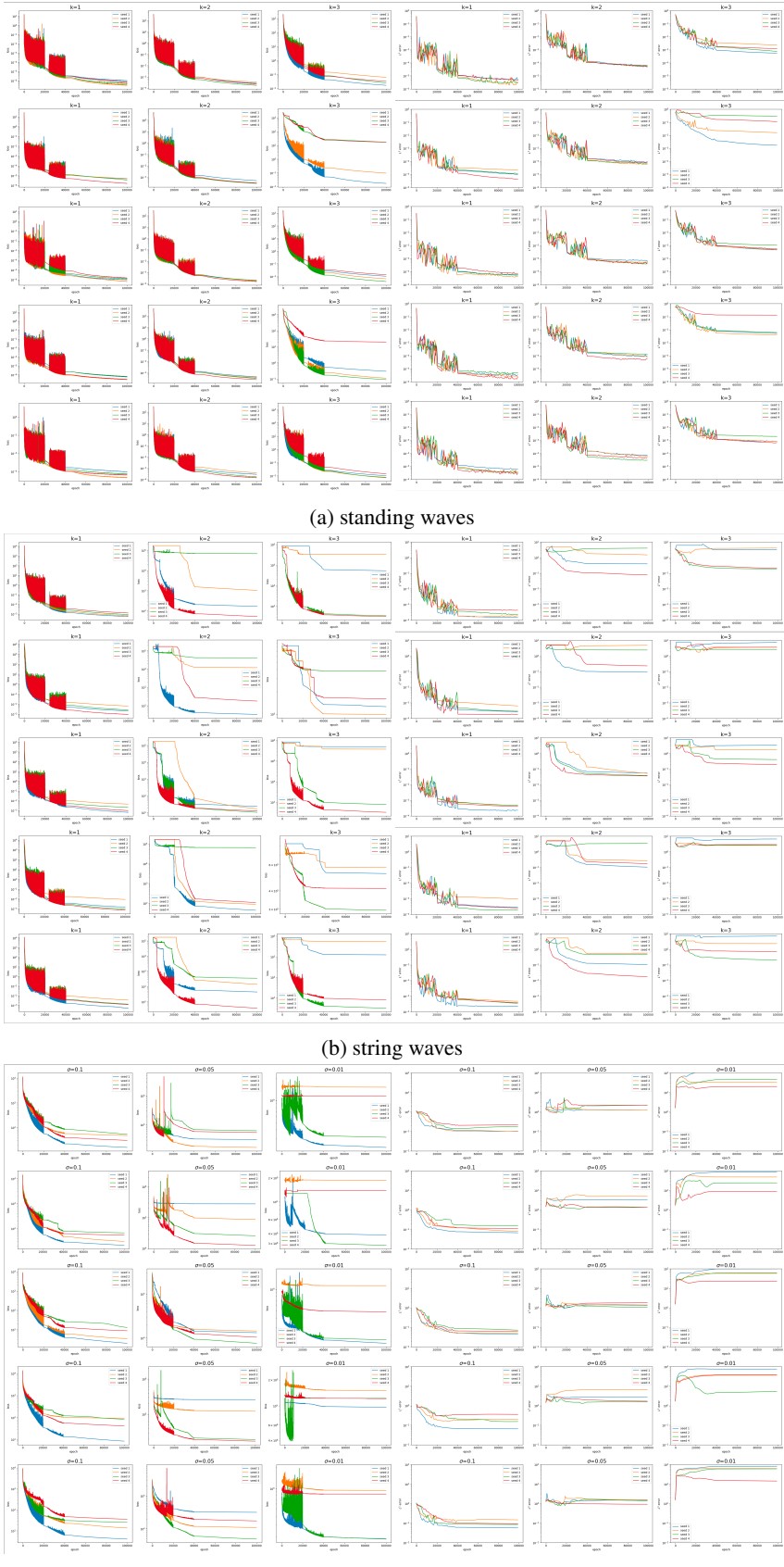

(a) standing waves

(b) string waves

(c) Gaussian traveling waves

Figure 9: $t^2, \frac{t^2}{t^2+1}, \frac{2t^2}{t^2+1}, \tanh^2(t), \left(\frac{\tanh(t)}{\tanh(1)}\right)^2$

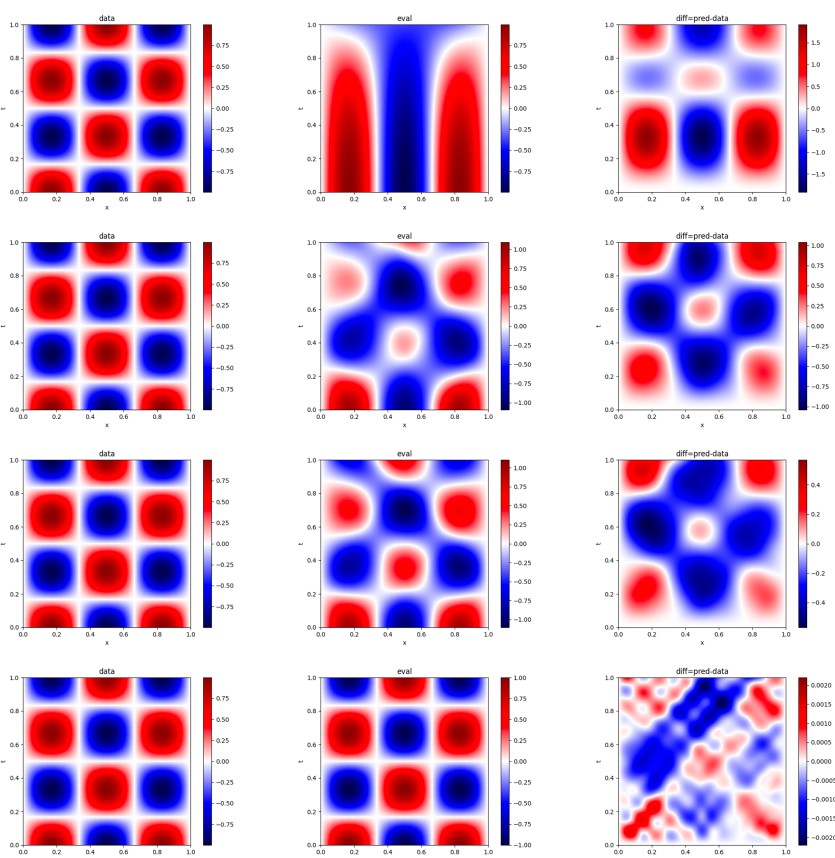

Figure 10: 0, 1000, 2000, and the last(converged)


Figure 11: 0, 10000, 20000, and the last(converged)

