# OpenReview forum: "Compute-Optimal Solutions for Acoustic Wave Equation Using Hard-Constraint PINNs"
_NeurIPS.cc/2024/Conference — Submitted to NeurIPS 2024_

### Official Review · Reviewer_AwP6 · 2024-06-21

**Soundness:** 2
**Presentation:** 1
**Contribution:** 2
**Rating:** 3
**Confidence:** 4

**Summary:**

The paper attempts to train PINNs which solves acoustic wave equations. They do so by using hard-constrained PINNs which can enforce IC and BCs, and propose a collocation point sampling method (DAFS) based on the amplitude of the solution at different regions.

**Strengths:**

The paper considers an interesting problem in acoustics and attempt to apply the techniques from PINNs to solve them.

**Weaknesses:**

The paper itself feels less coherent, and seems like just an application of many existing PINN training techniques (e.g., hard constraint PINNs, collocation point sampling) into solving a certain problem, rather than providing a novel method or a coherent framework into solving a domain-specific problem.

The experimental section feels incomplete. Different point selection algorithms have not been extensively compared with, e.g., from that in Wu et. al. (2023). Furthermore, it would be interesting to see how the method can scale to more realistic acoustic problems (i.e., outside of 1D settings).

The paper itself also seems incomplete. The Appendix and the NeurIPS checklist are partially filled and have half-finished sentences.

The labels within the graphs can also be enlarged slightly to make them more readable.

**Questions:**

1. Can the optimal \alpha be selected without having to test out different values of \alpha? Is there some recommended value that can be used for different acoustics problems?

2. Are there any relation between selecting points in high-amplitude regions with selecting points in high-residual regions, such as those methods benchmarked in Wu et. al. (2023)? In the sense that these two are indirectly doing the same thing, or they end up selecting very similar points anyway.

3. In Figure 5, are there any intuition to why L1 loss peaks at larger \alpha but L2 peaks at smaller \alpha?

4. How is the computation time of the methods proposed?

**Limitations:**

The authors have provided limitations with selection of \tau.

---

> ### Author Rebuttal · Authors · 2024-08-07
>
> We thank Reviewer \textbf{AwP6} for their constructive comments and appreciations of our strengths such as "The paper considers an interesting problem in acoustics".
>
> \issue{Weaknesses}
> % The paper itself feels less coherent, and seems like just an application of many existing PINN training techniques (e.g., hard constraint PINNs, collocation point sampling) into solving a certain problem, rather than providing a novel method or a coherent framework into solving a domain-specific problem.
>
> Our intention is to propose a general framework for imposing hard constraints into PINNs, particularly for acoustics problems with non-negligible first time derivative terms. Very few studies have thoroughly discussed the hard-constraint embedding of this first time derivative term.
>
> %The experimental section feels incomplete. Different point selection algorithms have not been extensively compared with, e.g., from that in Wu et. al. (2023). Furthermore, it would be interesting to see how the method can scale to more realistic acoustic problems (i.e., outside of 1D settings).
>
> We agree that the experimental section would benefit from a more extensive comparison with other point selection algorithms. In the revised manuscript, we will include a comparative analysis with methods such as those benchmarked in Wu et al. (2023). Additionally, we have expanded the scope of our experiments to include the application of our method to the 2D wave equation, demonstrating its scalability to more realistic acoustic problems. The results are detailed in the attached PDF.
>
> %The paper itself also seems incomplete. The Appendix and the NeurIPS checklist are partially filled and have half-finished sentences.
>
> We apologize for the incomplete sections in the original rushed submission. We will ensure that the Appendix and the NeurIPS checklist are fully completed in the revised version, with all sentences properly finished.
>
> %The labels within the graphs can also be enlarged slightly to make them more readable.
>
> We will enlarge the labels within the graphs in the revised version to enhance readability.
>
> \issue{Questions (1)}
> % Can the optimal \alpha be selected without having to test out different values of \alpha? Is there some recommended value that can be used for different acoustics problems?
>
> We can choose a medium $\alpha$ around $0.5$ based on our preliminary experiments. Alternatively, conducting a few pre-experiments can help in selecting the optimal $\alpha$ for different acoustic problems.
>
> \issue{Questions (2)}
> % Are there any relation between selecting points in high-amplitude regions with selecting points in high-residual regions, such as those methods benchmarked in Wu et. al. (2023)? In the sense that these two are indirectly doing the same thing, or they end up selecting very similar points anyway.
>
> This is an insightful question. Selecting points in high-amplitude regions relies on assumptions about the PDE solutions. However, selecting high-residual regions typically requires trial and error during training, whereas high-amplitude regions can be identified more directly based on the physical characteristics of the solution.
>
> \issue{Questions (3)}
> % In Figure 5, are there any intuition to why L1 loss peaks at larger \alpha but L2 peaks at smaller \alpha?
>
> This is an interesting observation. The difference in the behavior of L1 and L2 losses with respect to $\alpha$ might be due to their sensitivity to outliers and the distribution of error.
>
> \issue{Questions (4)}
> %How is the computation time of the methods proposed?
>
> The computation time of our methods is comparable to standard PINN training techniques. However, the inclusion of hard constraints and Dynamic Amplitude-Focused Sampling (DAFS) can slightly increase the computation time due to additional calculations. We will provide a detailed analysis of computation time in the revised manuscript.
>
> \issue{Limitations}
> % The authors have provided limitations with selection of \tau.
>
> We have included the limitations with selection of $\tau(t)$ in the original submission.

---

> > ### Comment · Reviewer_AwP6 · 2024-08-14
> >
> > I thank the author for their response. I still believe that the paper requires some amount of revision and therefore will keep the current score.

---

### Official Review · Reviewer_cEZM · 2024-07-03

**Soundness:** 2
**Presentation:** 2
**Contribution:** 1
**Rating:** 3
**Confidence:** 5

**Summary:**

The manuscript treats the one dimensional wave equation with a PINN approach and discusses the imposition of boundary and initial conditions directly into the network, as common practice in PINNs. The authors then propose a quadrature scheme based on a coarse finite difference discretization of the wave equation.

**Strengths:**

The imposition of the time derivative seems to be a novel construction. Furthermore, the construction seems not to be limited to the wave equation.

**Weaknesses:**

The main weakness of the manuscript is the focus on the very special and simple toy problem of the one dimensional wave equation. Solving the one-dimensional wave equation with PINNs is only of academic interest and insights obtained from it for the training of PINNs might not generalize. More specifically:
- The exact imposition of the time derivative should also work for general time dependent equations. The authors should comment on this.
- The sampling strategy employing a finite difference simulation to determine regions of high sampling density is not a generalizable approach. If a finite difference solver for the equation at hand is available, a PINN solver is typically not required.
- The authors determine an optimal function $\tau$ via considering six concrete examples. There is no guarantee that this approach will generalize to different equation types and is therefore of limited practical use.
- The authors might want to discuss the theoretical literature that proves the theoretical advantage of exactly imposed boundary conditions [1, 2, 3] and more elaborate constructions of distance functions.

[1] https://proceedings.mlr.press/v190/muller22b/muller22b.pdf

[2] https://arxiv.org/abs/2311.00529

[3] https://www.sciencedirect.com/science/article/abs/pii/S0045782521006186

**Questions:**

See weaknesses.

**Limitations:**

The scope of the paper is too narrow.

---

> ### Author Rebuttal · Authors · 2024-08-07
>
> We thank Reviewer \textbf{cEZM} pointing out. We acknowledge the concern about the focus on the one-dimensional wave equation. While the 1D wave equation serves as an initial validation, our intention is to propose a general framework for imposing hard constraints in PINNs, including for the first time derivative. To address this, we have added the application of our method to the 2D wave equation in the attached PDF, demonstrating its broader applicability.
>
> \issue{Weakness (1)}
> % The exact imposition of the time derivative should also work for general time dependent equations. The authors should comment on this.
>
> We thank Reviewer \textbf{cEZM} for pointing out the applicability of our method to general time-dependent equations. Indeed, the exact imposition of the time derivative is designed to be general and can be applied to a wide range of time-dependent PDEs. Compared to commonly benchmarked elliptic and parabolic partial differential equations, wave equations are more challenging to solve numerically.
>
> \issue{Weakness (2)}
> % The sampling strategy employing a finite difference simulation to determine regions of high sampling density is not a generalizable approach. If a finite difference solver for the equation at hand is available, a PINN solver is typically not required.
>
> We agree that if a finite difference solver is available, it might reduce the necessity of using PINNs. However, considering future incorporation with real noisy data, PINNs have advantages in unifying forward simulation and inverse problems.
>
> \issue{Weakness (3)}
> % The authors determine an optimal function via considering six concrete examples. There is no guarantee that this approach will generalize to different equation types and is therefore of limited practical use.
>
> We appreciate this observation. While our initial study explored six candidate functions to determine an optimal approach, we acknowledge the need for broader validation. In the revised version, we will discuss the limitations of this approach and suggest that future work should explore a more extensive set of functions and equation types to enhance generalizability.
>
> \issue{Weakness (4)}
> % The authors might want to discuss the theoretical literature that proves the theoretical advantage of exactly imposed boundary conditions [1, 2, 3] and more elaborate constructions of distance functions.
>
> We appreciate the papers shared by Reviewer \textbf{cEZM}. These papers will help us make a better discussion of the advantages of exactly imposed boundary conditions in our revised version.
>
> \issue{Questions}
> % See weaknesses.
>
> We have answered the questions in the weakness session.
>
> \issue{Limitations}
> % The scope of the paper is too narrow.
>
> We have added the application of this method to the 2D wave equation. The results are in the attached PDF.

---

> > ### Comment · Reviewer_cEZM · 2024-08-12
> >
> > Thank you for your answer. I think the pre-print still needs a decent amount of work,  I am still critical about relying on a finite difference solver within a PINN workflow. I would like to keep my score.

---

### Official Review · Reviewer_U3hW · 2024-07-11

**Soundness:** 2
**Presentation:** 2
**Contribution:** 1
**Rating:** 2
**Confidence:** 5

**Summary:**

This paper explores to solve the acoustic wave equation in the context of PINNs. Hard boundary and initial conditions are enforced by employing continuous functions within the PINN ansatz to ensure that these conditions are satisfied. A Dynamic Amplitude-Focused Sampling (DAFS) method is introduced to improve the efficiency of hard-constraint PINNs under a fixed number of sampling points.

**Strengths:**

1. Propose a general hard constraint imposition formula which correctly imposes all boundary conditions and initial conditions as required.

**Weaknesses:**

1. Only the wave equation is discussed.
2. The proposed Dynamic Amplitude-Focused Sampling (DAFS) method is trivial.
3. There are no comparisons with other methods in the experiments.
4. In the experiments, the relative errors between exact solutions and predictions are not given.
5. In the context of PINNs, it is better to give explicitly the formulation of training loss. Training details are also lacking.
6. Instead of tuning \tau (t) manually, it is better to train \tilde{u}(x,t) and \tau (t) simutanuously.
7. Many typos and grammar errors, such as "both and \alpha" in line 149, "x \in {\partial \Omega}_i" in line 125, "computational" in line 46.
8. The quality of Fig.7 should be improved.

**Questions:**

In line 41, what does the "basic function" mean? Is it the function \tau (t)?

**Limitations:**

Only the wave equation is discussed. There are no comparisons with other methods in the experiments.

---

> ### Author Rebuttal · Authors · 2024-08-07
>
> We thank Reviewer \textbf{U3hW} for their constructive comments and appreciation of our strengths, such as "The hard constraint imposition formula are general."
>
> \issue{Weaknesses (1)}
> % Only the wave equation is discussed.
> The wave equation is the focus of our study. We are proposing a general framework to impose hard constraints into PINNs, including the hard constraint for the first time derivative, and the wave equation is particularly suitable for this study. Additionally, compared to commonly benchmarked elliptic and parabolic partial differential equations, wave equations are more challenging to solve numerically.
>
> \issue{Weaknesses (2)}
> % The proposed Dynamic Amplitude-Focused Sampling (DAFS) method is trivial.
> The DAFS method effectively distributes samples to achieve the same level of accuracy with fewer samples compared to the vanilla uniform distribution. Furthermore, DAFS incurs minimal computational cost. However, more studies are needed to determine the optimal use cases for DAFS and how to choose the high- and low-amplitude regions effectively.
>
> \issue{Weaknesses (3)}
> % There are no comparisons with other methods in the experiments.
> We agree that comparing our approach with other methods would strengthen our study. In the original submission, we only showed the comparison of our sampling method with vanilla uniform random sampling. In the revised manuscript, we will include a comparative analysis with methods such as importance sampling and adaptive sampling to highlight the strengths and limitations of our approach.
>
> \issue{Weaknesses (4)}
> % In the experiments, the relative errors between exact solutions and predictions are not given.
> The relative errors between exact solutions and predictions are shown in Figures 10 and 11 in the Appendix. We apologize for not mentioning this in the main text.
>
> \issue{Weaknesses (5)}
> % In the context of PINNs, it is better to give explicitly the formulation of training loss. Training details are also lacking.
> We will add training details in the revised version. For our updated results of the 2D wave equation, we include the training details in the caption in the attached PDF.
>
> \issue{Weaknesses (6)}
> % Instead of tuning \tau (t) manually, it is better to train \tilde{u}(x,t) and \tau (t) simultaneously.
> This is an interesting direction. We will explore this in future work. In the original manuscript, we selected $\tau(t)$ from a set of candidate functions that can enforce the initial conditions.
>
> \issue{Weaknesses (7)}
> % Many typos and grammar errors, such as "both and \alpha" in line 149, "x \in {\partial \Omega}_i" in line 125, "computational" in line 46.
> Thank you for pointing that out. We apologize for these typos due to a rushed submission. We will correct all these typos in our revised version.
>
> \issue{Weaknesses (8)}
> % The quality of Fig.7 should be improved.
> We will redraw Fig. 7 in the revised version.
>
> \issue{Questions}
> % In line 41, what does the "basic function" mean? Is it the function \tau (t)?
> Yes. We will correct line 41 to ``(...) the basic function $\tau(t)$ (...)''.
>
> \issue{Limitations}
> % Only the wave equation is discussed. There are no comparisons with other methods in the experiments.
> In the original submission, we only showed the comparison of our sampling method with vanilla uniform random sampling. In the revised manuscript, we will include a comparative analysis with methods such as importance sampling and adaptive sampling to highlight the strengths and limitations of our approach.

---

> > ### Comment · Reviewer_U3hW · 2024-08-13
> >
> > Thank you for your answer.  Since some concerns have not been solved, I would like to keep my score.

---

### Official Review · Reviewer_QJWv · 2024-07-12

**Soundness:** 3
**Presentation:** 2
**Contribution:** 2
**Rating:** 3
**Confidence:** 3

**Summary:**

This paper improves the training efficiency of original physics-informed neural networks to solve the 1D wave equation threefold: first by extending ansatz to also take the first derivative into account, second by a sampling method that focuses on high-amplitude regions, and third by a framework for domain decomposition.

**Strengths:**

+ The related work is well presented.
+ The evaluation of the six candidate functions for \tau in section 4.2 provides interesting insights. The authors explore an advanced selection method for \tau based on the task at hand which might be an interesting research direction.

**Weaknesses:**

[Originality] While considering the first derivative for the ansatz is a good addition, the contribution is only minor.
Sampling more collocation points in the regions that might be more difficult to solve is a practical approach however the comparison and distinction to other sampling methods is missing.
Lastly if I understand the domain decomposition framework correctly, the contribution is to wrap the entire training into a loop and, based on the training process's results, increase or decrease the subdomain size.

Evaluation results are only provided for the 1D wave equation. Further results for other differential equations are necessary to demonstrate the benefits of the proposed method.

[Clarity]
The framework for domain decomposition is not presented clearly. While the flow chart in Figure 7 provides an overview of the method additional textual explanations in Section 4.4 are needed.
There were few to no remarks about the training regime (#training points, optimizer, learning rate…, etc.), making it more difficult to reproduce results.
Minor remarks:
-            N_pde is not introduced. It is probably the number of collocation points?
-            Most of the Figures (e.g. Fig. 1, Fig 6.) are hard to read.
-            Line 46: (…) optimal size of the computational [domain?] given (…)
-            Line 149: Both [N_pde?] and alpha (…)
-            Line 178: (…) In general, (...) performs better in general

**Questions:**

Q1: In the abstract the DAFS method is described as a method "that optimizes the efficiency (…) under a FIXED number of sampling points. However in Section 3.2: "This strategy optimally selects the number of points, N_pde, used in training." I assumed that DAFS only distributes the collection points into high and low-amplitude regions. Is that correct?
Q2: How does DAFS compare to other state-of-the-art collocation point sampling methods? When should one use DAFS, and in which cases a different method?

**Limitations:**

While the authors clearly state that they are interested in the 1D wave equation it would have been interesting to see their proposed methods applied to the 2D wave equation of any other differential equations what are typically used in PINN benchmarks.

---

> ### Author Rebuttal · Authors · 2024-08-07
>
> We thank Reviewer \textbf{QJWv} for their constructive comments and appreciations of our strengths such as "The related work is well presented" and "The evaluation of the six candidate functions for $\tau$ in section 4.2 provides interesting insights. The authors explore an advanced selection method for $\tau$ based on the task at hand which might be an interesting research direction".
>
> \issue{Weaknesses (Originality)}
>
> We thank Reviewer \textbf{QJWv} for the acknowledgment of our addition. While we recognize that considering the first derivative provides a modest improvement, this enhancement is crucial for wave equation systems with non-negligible first time derivative terms. Very few studies have thoroughly discussed the hard-constraint embedding of this first time derivative term.
>
> Thank you for the suggestion regarding the sampling method. We agree that comparing our approach with other sampling methods would strengthen our study. In the original submission, we only showed the combination of our sampling method with vanilla uniform random sampling. In the revised manuscript, we will include a comparative analysis with methods such as importance sampling and adaptive sampling to highlight the strengths and limitations of our approach.
>
> Your understanding of our domain decomposition framework is correct. This framework adapts the subdomain size dynamically based on the training process's results, which helps in efficiently handling complex regions. We apologize for not clarifying this process in the manuscript due to the rush. We hope we will have the opportunity to provide more detailed examples and results in the revised version to illustrate the benefits of this adaptive approach.
>
> \issue{Weaknesses (Clarity)}
>
> We thank Reviewer \textbf{QJWv} for clearly pointing out the clarity issues in our paper. The initial submission was rushed, resulting in insufficient explanations in Section 4.4, typos, and less polished figures.
>
> Thank you for kindly highlighting these minor clarity problems. We will address the \textbf{Minor remarks} as follows:
>
> 1. Yes, N\_pde refers to the number of collocation points used to calculate the PDE loss in PINNs.
>
> 2. We have updated Fig. 1 and Fig. 6. Fig. 1 shows the ground truth results of the benchmark used in this paper, where the x-axis represents the spatial variable $x$ and the y-axis represents the time variable $t$. Fig. 6 illustrates the different masking rates of the sampling strategy, and the results in Fig. 5 indicate that the best masking rate $\alpha$ is around 0.5.
>
> 3. Line 46: We will correct this to ``optimal size of the computational domain given (...)''.
>
> 4. Line 149: We will correct this to ``N\_pde and alpha (…) ''.
>
> 5. Line.178: This is a typo; we mean that $\tau=t^2,2t^2/(1+t^2)$ generally performs better.
>
> \issue{Questions (Q1)}
>
> Thank you for this question. We apologize for the confusion. You are correct that DAFS distributes the collocation points into high and low-amplitude regions in the example. In Section 3.2, we intended to convey that the redistribution of samples has the potential to achieve the same level of accuracy with fewer samples compared to the vanilla uniform distribution. However, we do not directly select the number of points, N\_pde, used in training. We will correct the sentence to "This strategy optimally distributes collocation points in training."
>
> \issue{Questions (Q2)}
>
> Thank you for this question. In our original submission, we only compared DAFS with the vanilla random sampling method. We will work on providing a more comprehensive comparison in our revised version.
>
> From the observation of our numerical experiments on the wave equation, DAFS outperforms the random sampling method for traveling waves but underperforms for standing waves. This can be seen in Figure 2 of the attached PDF.
>
> \issue{Limitations}
>
> In Figure 1 of the attached PDF file, we have added the results of the 2D wave equation with $\tau(t) = t^2$. The comparison of ground truth and PINN-predicted wave fields of standing 2D waves shows that the hard-constraint embedding guarantees accuracy at the boundaries and initial steps but struggles to scale to large time domains when the frequency is high.

---

### Author Rebuttal · Authors · 2024-08-07

We thank Reviewer \textbf{QJWv}, \textbf{U3hW}, \textbf{cEZM} and \textbf{AwP6} for their constructive comments and appreciations of our strengths such as "The related work is well presented", "The evaluation of the six candidate functions for $\tau$ in section 4.2 provides interesting insights. The authors explore an advanced selection method for $\tau$ based on the task at hand which might be an interesting research direction", "The hard constraint imposition formula are general", "The imposition of the time derivative seems to be a novel construction", "The construction seems not to be limited to the wave equation" and "The paper considers an interesting problem in acoustics".

We answer each reviewer's questions separately under their reviews. Thank you all again for reviewing our abstract! There are so many insightful questions and suggestions that can help us improve our abstract.

---

### Decision · Program_Chairs · 2024-09-25

**Decision:**

Reject

**Comment:**

This paper proposes to improve the training efficiency of physics-informed neural networks, first by extending ansatz to also take the first derivative into account, second by a sampling method that focuses on high-amplitude regions, and third by a framework for domain decomposition. The reviewers generally think that, in the current state, the scope of the paper is quite narrow as it only tackles the 1D wave equation, the paper lacks comparisons with existing approaches, and the presentation needs to be significantly improved to avoid confusions.